# Chemokine Heteromers and Their Impact on Cellular Function—A Conceptual Framework

**DOI:** 10.3390/ijms241310925

**Published:** 2023-06-30

**Authors:** Xavier Blanchet, Christian Weber, Philipp von Hundelshausen

**Affiliations:** 1Institute for Cardiovascular Prevention (IPEK), LMU Munich, 80336 Munich, Germany; xavier.blanchet@med.uni-muenchen.de (X.B.); christian.weber@med.uni-muenchen.de (C.W.); 2German Centre for Cardiovascular Research (DZHK), Partner Site Munich Heart Alliance, 80636 Munich, Germany; 3Department of Biochemistry, Cardiovascular Research Institute Maastricht (CARIM), Maastricht University Medical Centre, 6229 HX Maastricht, The Netherlands

**Keywords:** chemokine, homodimer, heterodimer, protein–protein interactions, galectin, therapeutics

## Abstract

Chemoattractant cytokines or chemokines are proteins involved in numerous biological activities. Their essential role consists of the formation of gradient and (immune) cell recruitment. Chemokine biology and its related signaling system is more complex than simple ligand–receptor interactions. Beside interactions with their cognate and/or atypical chemokine receptors, and glycosaminoglycans (GAGs), chemokines form complexes with themselves as homo-oligomers, heteromers and also with other soluble effector proteins, including the atypical chemokine MIF, carbohydrate-binding proteins (galectins), damage-associated molecular patterns (DAMPs) or with chemokine-binding proteins such as evasins. Likewise, nucleic acids have been described as binding targets for the tetrameric form of CXCL4. The dynamic balance between monomeric and dimeric structures, as well as interactions with GAGs, modulate the concentrations of free chemokines available along with the nature of the gradient. Dimerization of chemokines changes the canonical monomeric fold into two main dimeric structures, namely CC- and CXC-type dimers. Recent studies highlighted that chemokine dimer formation is a frequent event that could occur under pathophysiological conditions. The structural changes dictated by chemokine dimerization confer additional biological activities, e.g., biased signaling. The present review will provide a short overview of the known functionality of chemokines together with the consequences of the interactions engaged by the chemokines with other proteins. Finally, we will present potential therapeutic tools targeting the chemokine multimeric structures that could modulate their biological functions.

## 1. Introduction

Proteins are one of the four major classes of biological macromolecules and interactions between them, protein–protein interactions (PPIs), are one of the central pillars that are universally required in regulating biological processes such as cell signaling and metabolism, macromolecular assembly, sensing or vascular integrity [1]. However, the human reference interactome mapping project HuRI built from yeast two hybrid screens comprising >53,000 interactions does not detect any heteromeric interactions of soluble mediators reflecting a current necessity to study these PPI selectively [2]. In contrast to the interactions of intracellular and membrane proteins, the requirement of heterophilic interactions for activity between soluble effector proteins has been shown only for a few examples. The pituitary peptide hormones TSH, FSH and LH share the glycoprotein hormone alpha chain (CGH), which forms by non-covalent association with the unique β-subunit active heterodimers [3]. Only as a pair do they bind and activate their G-protein-coupled receptors. Another example for a successful translation of the finding that cytokines can form functional important heterodimers is IL17A•IL17F. It has been shown that IL-17A and IL-17F, two distinct cytokines that are released by Th17 cells, are associated with heterodimers that facilitate signaling by heterodimerization of IL17 receptors A and C [4]. This is important for inflammatory mechanisms, and binding this heterodimer through the bifunctional antibody bimecizumab is now an approved therapy in psoriasis and shows beneficial effects in psoriasis arthritis [5]. In addition, the cytokines of the IL-12 family, which includes IL-12, IL-23, IL-27 and IL-35, form non-covalent heterodimers by pairing α subunits (p19, p28 or p35) and β subunits (p40 or Ebi3) with contrasting biological effects in inflammation [6]. However, subunits can exert some activities on their own or as homodimers, as shown for the p40 subunit of IL-12 [7]. The heterodimerization of cytokine subunits mirrors and facilitates the signaling through heterodimeric receptors that are composed of subunits IL12Rβ1, IL12Rβ2, IL6ST and IL27RA.

Chemokines are chemotactic cytokines that represent a family of soluble signaling molecules of approximately 70 amino acid residues with a molecular weight of 7–12 kDa [8]. Moreover, the sequence identities between human chemokines can greatly vary from less than 20% to over 90%. Chemokines are involved in numerous biological processes under physiological conditions such as homeostasis, development, tissue repair and angiogenesis, but also under pathological disorders including tissue damage, bacterial or viral infection, tumorigenesis, cancer metastasis and inflammatory and autoimmune diseases.

To date, approximately 50 human chemokines have been identified and classified into one of four subfamilies, CXC, CC, CX3C and XC, based on the arrangement of cysteine residues involved in the formation of disulfide bonds. The CC- chemokines are the largest sub-family (or β-family), followed by the CXC chemokines (α-family). The C chemokine (δ-) sub-family consists of only two members (XCL1/2). Fractalkine is the only member of the CX3C sub-family (γ-chemokine). It has an intracellular cytoplasmic domain and is anchored to the cell membrane through a transmembrane region. Another family has been described in the zebrafish genome, namely the CX family, which lacks one of the four cysteine residues highly conserved amongst chemokines [9].

## 2. General Structure of Chemokine Monomers

Chemokines are produced as pro-peptides of around 100 amino acids. During its processing, a signal peptide of approximately 20 residues is cleaved, leading to the mature form of the protein. Chemokines fold into a common tertiary structure despite their high sequence variability [10]. The first two cysteines are found together near the flexible N-terminal end of the mature protein, the third cysteine is localized in the center of the molecule and the fourth is close to the C-terminal end. A loop of approximately ten amino acids follows the first two cysteines and is known as the N-loop. A right-handed helical structure, called a 3_10_-helix, follows the N-loop, followed by three β-strands and a C-terminal α-helix. These helices and strands are connected by turns called 30s, 40s and 50s loops, based on the numbering of residues in the mature protein; the third and fourth cysteines are thus located in the 30s and 50s loops, respectively (Figure 1).

Interestingly, when murine CXCL13, a chemokine with 19 residues following the C-terminal helix, harbors an N-terminal methionine, which occurs when the protein is recombinantly expressed in *E. coli*, the tertiary fold adopts a new conformation where the N-terminal sequence folds over the first β-strand of the chemokine [11]. This dramatic change would prevent its dimerization and receptor activation. This finding highlights the importance of using mature forms of chemokines when studying their structure-function relationship.

The human chemokine XCL1 is a metamorphic protein that exists under two native conformations at near-physiologic conditions, a canonical α-β chemokine fold, and a four-stranded β-sheet that forms a dimer [12]. XCL1 swaps spontaneously and reversibly between the two structures, and this interconversion dictates the chemokine activity [13].

## 3. Chemokine Homodimerization and Function

Structural analysis has revealed that chemokines exist as monomers but can also form dimers or even higher oligomeric structures. For instance, CXCL8 or CXCL12 are found as both a monomer and a dimer in solution [14,15]. The formation of oligomers is strongly influenced by the amino acid composition and conformation of the interfaces of the intersubunits, thereby determining the thermodynamic stability of the complexes [16]. Two dimeric conformation types have been observed. In a CC-type, dimerization occurs via interaction of the flexible N termini to form a two-stranded antiparallel β-sheet, whereas in a CXC-type, chemokines interact by antiparallel extension of preformed β-strands. Some chemokines, namely CCL2, CCL27 and CXCL12, have been identified in both conformations [17,18,19]. As mentioned before, the metamorphic chemokine XCL1 has an unusual and unique dimeric structure with distinct functional properties. Indeed, its dimer forms an all-β conformation (called Ltn40), which self-associates as a head-to-tail dimer with no C-terminal alpha helix fold, promoting GAG binding [12]. However, this high-affinity GAG-binding beta-sandwiched XCL1 dimer has lost the ability to activate XCR1 [20].

Chemokine concentration and/or physicochemical conditions of the microenvironment dictate the quaternary structure of the chemokines. For example, CXCL4 can exist as a monomer, dimer or tetramer depending on the pH and salt concentration, thereby influencing its biological activity via binding modulation to CXCR3A [21]. Likewise, the ternary structure of CCL28 appears to be sensitive to changes in pH and salt across a physiological range [22]. Thus, below pH 6.0 and with no salt, CCL28 lost its conformation along with its ability to bind and activate its cognate receptor, but preserved its antifungal activity. At pH above 6.0, the structure of CCL28 is maintained, allowing both binding and activation of the chemokine receptor. However, the antifungal activity of CCL28 disappeared if salt was also present in the milieu.

Oligomerization of CXCL7, one of the most abundant chemokines expressed by platelets and released during platelet activation, depends on the chemical environment [23,24]. Like CXCL4, it has been shown to exist as monomers, dimers and tetramers. Increasing the pH from 3.5 to 6 increases the association equilibrium for dimer formation of all CXCL7 species depending on the length of the N-terminus (PPBP > CTAPIII > β-TG > NAP-2), while that of the association to tetramers is decreased [23,24]. Ionic strength also plays a role, but effects are smaller and are not linear. Effects are due to modification of the global charge of some residues leading to the disruption of intramolecular electrostatic interactions. On the other hand, these modifications allow the generation of new electrostatic interactions with other proteins or the formation of salt bridges with ions from the milieu [24]. Then, these changes would foster the formation of new oligomeric states, which would promote or alter the chemokine gradient formation, the generation of a new heterodimer, and influence the receptor binding activity (e.g., CXCR2), which would eventually modulate the cellular/immune response (e.g., neutrophil recruitment to the site of injury).

Likewise, native spray mass spectrum of wild-type CCL5 at high concentration (10µM) and low pH (4.5) has revealed a higher oligomeric state consisting of concatenation of a dimer sub-structure leading to an octameric state [25]. Interestingly, covalent tyrosine–tyrosine cross-linking multimeric forms of CCL5 were obtained using copper/H2O2 redox system at non-cytotoxic concentrations [26]. The cross-linked oligomeric form is highly resistant to proteolysis. Furthermore, heparan sulphate acted as an anti-oxidant, limiting multimerization at low concentrations and protecting high molecular weight multimers at high concentrations of hydrogen peroxide. These findings demonstrated that the physicochemical properties of the microenvironment surrounding chemokines greatly affect their structures, thereby tailoring their biological functionalities.

One characteristic of chemokines is their ability to bind GAGs, and oligomerization appears to be critical for the GAG-binding affinity of many chemokines; in some cases, this confers GAG specificity [27]. Structurally, distinct heparin-binding motifs have been identified in several CC- and CXC-chemokines. Many CC chemokines bind through a basic BBXB motif, where B represents a basic amino acid residue that is located in the 40s loop of CC chemokines [28,29]. The type of GAG encountered could also reveal different and additional GAG binding sites. Accordingly, chondroitin sulfate fragments revealed that CCL5 dimers have another GAG-binding epitope localized at the N loop of the protein, a region involved when binding to the cognate receptor [30]. Numerous heparin-binding sites have been defined in CXC chemokines. In the case of CXCL4, binding occurs at the C-terminus of the protein, whereas in CXCL11, the 50 loop is responsible for polysaccharide binding [31,32]. CXCL1, which like CXCL8 has the ELR motif critical for the activation of the chemokine receptor CXCR2, has two distinct GAG-binding domains. The N-loop and C-helical residues constitute the α-domain and are similar to the GAG-binding domain of CXCL8. The N-terminal, 40s turn and third β-strand residues define the β-domain and are located on the opposite side of the protein [33]. The authors of the study proposed two models of CXCL1 and CXCL8 binding to heparan sulfate: the “clamp” and “horseshoe” models. In the former prototype, a dimer of CXCL1 is sandwiched between the sulfated regions of heparan sulfate. The latter model suggested that the N-acetylated domain of heparan sulfate would bridge the two α-helices, whereas the two adjacent sulfated regions would each run alongside one helix [34].

Furthermore, structure experiments reveal that the oligomerization of chemokines affects how GAGs bind chemokines. CCL3, CCL4 and CCL5 have a common and pivotal GAG-binding motif in the 40s loop that becomes partially buried at the moment these CC chemokines associate to higher order oligomers. While the CCL5 polymer binds GAGs through an additional motif in the 50s loops [35], CCL3 polymers allow GAG binding through novel binding sites that develop by proximity of two partially buried BBXB motifs [36]. The addition of GAGs might contribute to and promote the formation of higher order oligomer in a two-step mechanism that includes dimerization and positive cooperativity as the residues of the BBXB motif are solvent accessible and aligned along the oligomer surface, thus allowing binding of GAG chains [25,37]. Similarly, heparin crosses the entire six-stranded β-sheet of CXCL12 created by dimerization without involving the N terminus parts of the chemokines [38]. CXCL7, which forms a weak dimer but heterodimers with CXCL1, was found to have the monomer predominantly in the free form and the dimer in the GAG-bound form [39,40]. Thus, investigating heparin binding of NAP-2/CXCL7, Brown et al. showed that heparin exerts effects on multiple NAP-2/CXCL7 residues so that not one single binding model can harmonize all chemical shift perturbations, indicating that the GAG interface is plastic [23]. In addition, GAG-binding sites appear to overlap residues responsible for binding to the N-terminus of CXCR2 so that NAP-2/CXCL7 monomers bound to GAGs would not be able to fully bind CXCR2. These findings highlight that the interaction between chemokines and GAGs determine the generation of a chemokine gradient and receptor activation by setting the proportion of free to GAG-bound chemokines.

By binding to GAGs, oligomerization of chemokines promotes the formation of a local gradient by concentrating chemokines at specific sites and presenting them to (immune) cells [41]. Oligomerization also protects against proteolytic degradation, extending the half-life of the gradient. This is consistent with the inherent role of the chemokine, but should not only be limited to this aspect. While monomeric chemokines are able to bind and activate their receptor, suggesting that this conformation is necessary to retain full receptor activity, chemokine homodimers display mixed functions [42,43,44,45]. Disulfide-linked dimeric forms of the CC-type chemokines are unable to bind or activate their cognate receptors because the linked N-termini are unable to enter the receptor binding pocket [46,47]. CXC-type dimerization may provide contacts to the chemokine receptor that are absent in the monomeric form, suggesting that oligomers interact differently with the receptor [48,49]. Compared to the monomeric form, the obligate (or trapped) homodimer of CXCL8 binds CXCR2 with lower affinity [50]. For receptor phosphorylation and internalization, the dimer is as active as the monomer, but endocytosis is reduced, suggesting fine-tuned activity and regulation of cellular function. In contrast, the monomer bound CXCR1 with higher affinity and was more active [50]. Conversely, the trapped CXCL1 dimer exhibited a strong agonism for CXCR2, which was comparable to the monomer [51].

Recently, a new functionality of chemokine dimerization has emerged, as shown for dimeric CXCL12. In addition to the cognate receptor CXCR4, CXCL12 homodimers bind to ACKR1 orders of magnitude stronger than in their monomeric form and promote ACKR1 internalization [52].

Although the relationship between chemokines and cancer has been extensively studied, little is known about the effects of chemokine dimerization. CXCL12 interfered with colon carcinoma and melanoma tumor metastasis via distinct interactions with CXCR4 [49]. Thus, while monomeric CXCL12 inhibited cAMP signaling, recruited β-arrestin-2, mobilized intracellular calcium and stimulated cell migration, dimeric CXCL12 activated G-protein-dependent calcium flux, adenylyl cyclase inhibition and rapid activation of ERK1/2.

CXCL4, as a tetramer, has been shown to transduce signals in endothelial cells and neutrophils via interaction with heparan sulfate or chondroitin sulfate, respectively [41,53,54]. This signaling controlled the phosphorylation of proteins that mediate leukocyte adhesion to endothelial cells and endothelial permeability [53]. In the case of neutrophils, adhesion was affected by src-kinases, whereas CXCL4-mediated release of secretory products from neutrophils required further activation of p38 MAP kinase and phosphatidylinositol 3-kinase [55].

Lande et al. highlighted a novel type of interaction of CXCL4. Tetrameric CXCL4 formed complexes with both bacterial and human DNA [56]. Thus, CXCL4 organizes DNA into liquid–crystalline columnar immune complexes. Those complexes, which are present in blood and tissues from patients suffering from systemic sclerosis (SSc), are DNA-size-dependent and activate plasmacytoid dendritic cells in a TLR9-dependent but CXCR3-independent manner. Likewise, a recent study unveiled that CXCL4 formed complexes with RNA, thereby preventing its degradation, increased the maturation of IFN-I-primed myeloid dendritic cells, stimulated the production of pro-inflammatory cytokines in monocyte-derived dendritic cells and correlated with both IFN-I and TNF-α in the blood of SSc patients [57]. Of note, the uptake and translocation of CXCL4 to the nucleus appears to be an active mechanism. This process is dependent on the G-protein-coupled receptor, but GAG-independent [58].

## 4. Chemokine Heterodimerization Activity

Conceivably, chemokines are also able to form heterodimers. Similar to the chemokine homodimer, the heterodimer adopts different conformations, namely CC-, CXC- and mixed-type. Binding to GAG can stabilize chemokine heterodimers, as shown with Arixtra, a heparin pentasaccharide, which promoted the formation of heterodimers of CCL8•CCL11 and CCL2•CCL11 [59]. In silico simulations showed that both chemokine concentrations and physicochemical properties of the microenvironment could be used to predict dimer disruption, and that the presence or absence of specific residues at the interface of each subunit of the heterodimer determined the type of conformation. [16]. For instance, under specific circumstances (e.g., inflammation or wound healing), local release or scavenging of chemokines might shift the association balance between the different entities, favoring the formation of a particular dimeric type that might trigger new biological responses.

The chemokine heterodimer CCL3•CCL4 was the first description of a naturally occurring heterodimer that is actively released by activated human monocytes and peripheral blood lymphocytes [60]. Since then, an important body of work has identified multiple heterodimers. In a systematic approach, a comprehensive map of heterodimeric pairs was generated by screening of all possible human heterodimeric chemokine interactions (the chemokine interactome), identifying over 200 heterodimeric chemokine pairs. Recently, pairs with atypical chemokines including MIF have also expanded this list [61,62].

As with homodimers, binding between GAGs and heterodimers appears to play an important role. Thus, molecular docking based on NMR experiments combining the obligate heterodimer CXCL1•CXCL2 showed that asymmetry caused by chemokine heterodimerization affected binding to GAGs, and that whether GAGs are soluble or immobilized is of great importance [63].

Heterodimer formation appears to be selective. Pro-inflammatory and non-homeostatic chemokines are more likely to undergo heterologous interactions [61]. In inflamed skin, co-expression of CXCL10 and CCL22 allowed their physical interaction with each other through the first β-strand of CCL22 and thereby enhanced CCR4-mediated T cell migration via CCR4 in a CXCR3- and GAG-binding-independent manner [64]. Homeostatic chemokines seemed to be less interactive, although several pairs have been identified previously [65,66]. Therefore, formation of heteromers may have been developed as a regulatory function, particularly under inflammatory conditions.

CC-type interactions appeared to have synergistic effects, whereas CXC-type interactions showed to be inhibitory. In vitro, the combination of CCL5 with CXCL12 (CXC-type) resulted in a perturbation of residues that are in line with CXC heterodimerization. Functionally, the addition of CCL5 inhibited CXCL12-induced chemotaxis of activated human T cells and hampered the stimulatory effect of CXCL12 on platelets in a non-competitive manner. Ex vivo, CXCL12 facilitated human and mouse platelet aggregation by activating CXCR4 and downstream PI3K, Syk and Btk signaling [67]. This effect was abolished by the formation of a CCL5•CXCL12 heterodimer. In DSS colitis, synergistic effects of mesenchymal stem-cell-derived CCL2 and CXCL12 were attributable to CCL2•CXCL12 heteromers that induced peritoneal-cavity-derived macrophage M2 polarization along with increased expression and the release of IL-10 in a CCR2- but not CXCR4-dependant manner, leading to the mitigation of intestinal inflammation. Interestingly, treatment with CCL2 alone was unable to upregulate IL-10 expression [68,69].

Heterophilic interactions between CXCL4 and CCL5 promote neutrophil recruitment in an LPS-induced acute lung injury model and accelerate atherosclerosis by triggering monocyte arrest on the endothelium [70,71,72]. Mechanistically, the GAG-binding capacity of the CCL5•CXCL4 heterodimer anchored the CCR1 receptor at the endothelial cell surface, thereby promoting the recruitment of immune cells. Contrary to CXCL4, combined stimulation of CXCL4L1 and CCL5 heterodimer failed to enhance the leukocyte recruitment both in vitro and in an in vivo peritoneal recruitment assay [73]. This could explain the decrease observed in atherosclerotic lesion size and macrophage content in a mouse model deficient for Cxcl4 and knocking in a mouse variant of human CXCL4L1 [61]. The atypical chemokine MIF, which structurally resembles the CXCL8 homodimer, forms an inhibitory heterocomplex with CXCL4L1 (but not with CXCL4) that attenuates MIF-mediated atherogenic and inflammatory activities by impairing the CXCR4 receptor axis. This blocks immune cell migration, endothelial adhesion and thrombus formation [62].

A fundamental role of the formation of chemokine heterodimer would be the simultaneous heterodimerisation and activation of two different chemokine receptors, as exemplified with CCL5 and CCL17 [61]. In vitro, both chemokines acted synergistically to enhance chemotaxis of interleukin-2-activated and CD3/C28-activated human T cells, involving both the CCL17 receptor (CCR4) and CCL5 receptors (CCR1 or CCR5). In vivo, CCL5•CCL17 heteromers were detected in aortic root lesions of *Apoe*^−/−^ mice where they impaired Tregs homeostasis by promoting CCR4•CCR5 heterodimerisation.

Early studies showed that CXCL4 bound to CXCL8 with high affinity and impaired CXCL8-dependant signaling in CD34+ human hematopietic progenitors, potentiated antiproliferative activity of CXCL4 against endothelial cells and enhanced chemotactic response to CXCL8 from CXCR2-transfected Baf3 cells [74,75]. Mixtures of CXCL4 and CXCL12 or obligate CXCL4•CXCL12 heterodimers inhibited CXCL12-induced migration of breast MDA-MB-231 cancer cells, increased Ca^2+^ release and activated downstream signaling of CXCR4 but not of CXCR3, highlighting its role in the limitation of cancer progression [76,77]. Furthermore, incubation of the chemokine mixture restored the expression of genes that have been associated with a decrease in overall survival in breast cancer patients. In those instances, formation of a heterodimer appeared to allow chemokine receptors to couple differently to downstream signaling pathways, leading to alternative cellular responses.

Because the tumor microenvironment can express multiple soluble factors including chemokines, it is highly likely that heterocomplexes can be formed as well [78]. For instance, CXCL9 and CXCL12 are expressed in the perivascular space of primary central system lymphoma that consists mainly of CD8 T cells. Co-existence of both chemokines resulted in CXCL9•CXCL12 complexes that enhanced the recruitment of B cells that lack the CXCL9/10/11 receptor CXCR3. To exploit this mechanism therapeutically and provide proof of causality, specific inhibitors of the CXCL12•CXCL9 interaction would be beneficial [79].

Using a trapped/obligate CXCL1•CXCL7 heterodimer, Brown et al. showed that the heterocomplex residues involved in GAG binding were different from their respective monomers [40]. As for monomeric CXCL1 and CXCL7, heterodimer–GAG interactions blocked the residues engaged in receptor binding, indicating that GAG-bound CXCL1•CXCL7 is unlikely to activate the receptor. Furthermore, CXCR2 receptor binding activities showed no differences between CXCL1, CXCL7, a mixture of both chemokines (i.e., native heterodimer) and the trapped heterodimer. Therefore, receptor activity and functional activities could be spatially and temporally controlled by GAGs by modulating the number of free chemokines.

Undoubtedly, due to the high number of heterodimeric pairs identified, the mechanisms described above might represent only part of them (Table 1). Recent data indicate that heterophilic dimeric formation with chemokines would not be as unbending as previously supposed [80]. The higher degree of flexibility of the dimer would improve the binding of and activation to (other) chemokine receptors, thereby potentiating the synergistic effect. Likewise, inhibitory effects of dimerization could also be related to structural modifications. Thus, CC-type dimer could provide more stable ß-sheet interactions, thereby attenuating monomer release, whereas α-helices alignment could alter CXC-type heteroformation.

Heterodimerisation could act as a switch mechanism to favor a dimeric type in order to (de)activate chemokine receptors for which activation was restricted previously. In solution, CCL2 forms a CC-type homodimer that is unable to bind to CCR2 because both N-termini involved in receptor binding and homodimerization are overlapping [17]. Posttranslational modifications including tyrosine sulfation are important for proper chemokine–chemokine receptor interactions as well. A sulphated peptide corresponding to the N-terminus of CCR2 binds tightly to CCL2 and promotes the dissociation of the CCL2 homodimer. However, tyrosine sulfation of chemokine receptor is unlikely to be complete and homogenous due to the differential activity of tyrosylprotein sulfotransferase in chemokine-receptor-expressing cells; thus, is unclear to which extent these findings can be translated into a physiologic context [81,82,83]. Because chemokines can form mixed types of heteromers [16,61], it is possible that heteromers of CC-chemokines, such as CCL2, can form CXC-type heterodimers, most likely with a CXC chemokine. This would result in an exchange of chemokine subunits so that a CC chemokine may bind and activate its receptor as a CXC type heteromer (Figure 2).

All of these mechanisms have yet to be demonstrated. However, it is reasonable to assume that they could occur in vivo.

Although numerous studies have highlighted the role of chemokine heterodimers in cellular function, some might present limitations. The major critical aspect is the use of individual chemokines. Indeed, the combination of different concentrations of chemokines will result in a solution containing a mixture of different oligomeric states, with monomeric, homo-/heterodimeric and multimeric structures. Whether the observed amplified or synergistic effects could also explain intracellular effects at the signaling level remains speculative. Trapped chemokine heterocomplexes appear to be the most prominent solution to overcome the issues because only one entity is present in the solution and its concentration can be fine-tuned to accurately control the experimental conditions. However, the construction of such proteins requires structural data that are often not available, rendering its design challenging. So far, two methods have been used to generate obligate/trapped heterodimers according to structural models derived from HSQC spectra and MD simulations. CXC-type heterodimers have been generated by the mutation of opposite relevant residues at the interface to cyteines and subsequent disulfide bridge formation [39,63]. An alternative approach to create obligate CC-type heterodimers has been chemical synthesis with oxime-ligation of opposite residues in the N-terminal region [84].

**Table 1 ijms-24-10925-t001:** Impact of (heterophilic) chemokine complexes on function-proposed mechanisms.

Dimer	Function	Cellular Effect	References
**Enhancing effects**			
CXCL4●CCL5	Prevention of CCR5 internalization, prolonged signaling	Increased monocyte arrest	[61,71,85,86]
CCL5●HNP1	Binding to CCR5	Increased recruitment and adhesion of monocyte, and macrophage recruitment	[87]
CCL5●CCL17	Promotion of CCR4•CCR5 heterodimerization	Promotes Treg homeostasis	[61]
CXCL10●CCL22	Activation of CCR4, increased ERK phosphorylation	Increased T cell migration	[64]
CCL2●CXCL12	Activation of CCR2 on macrophages, increased IL-10 production and secretion	M2 macrophage polarization, IL-10-dependant B and T cell activation	[68,69]
CXCL12●HMGB1	Ternary CXCR4 complex, biased agonism, β-arrestin 1/2-recruitment, actin polymerization, receptor internalization and degradation, increased intracellular Ca^2+^ mobilization and ERK phosphorylation	Increased leukocyte migration	[88,89]
CXCL12●CXCL12	Binding to and internalization of ACKR1	Binding to transfected MDCK cells and primary human Duffy-positive erythrocytes	[52]
CXCL4●Gal1	Modulation of the galectin glycan-binding affinity and specificity	Increased apoptotic activity of CD3+ and CD3+CD8+ T cells	[90]
**Inhibitory effects**			
CCL5●CXCL12	Biased CXCR4 antagonism, signaling through PI3K, Syk and BTK affected	Inhibition of platelet aggregation	[61,67]
CXCL4L1●MIF	Prevention of CXCR4 activation	Inhibition of MIF-stimulated thrombus formation and T cell migration	[62]
CXCL8●CXCL4	No experimental data addressing the molecular mechanism	Prevents CXCL8-induced activation of hematopietic progenitors	[74,75]
CXCL4●CXCL12	Ternary CXCR4 complex, biased antagonism, increased Ca^2+^ release	Suppression of MDA-MB 231 breast cancer cell migration	[76,77]
CXCL1●Evasin3	Absence of binding to CXCR2	Reduction of neutrophil migration	[91]
CXCL12●Gal3	Ternary complex with CXCR4, biased antagonism, decreased β-arrestin recruitment and cAMP production	Inhibition of CD4+ T cell, monocyte and neutrophil recruitment	[92]
CCL5●Gal9	Modulation of the galectin glycan-binding affinity andspecificity	Decreased apoptotic activity of CD3+ and CD3+CD4+ T cells	[90]
CXCL12●CXCL12	Biased CXCR4 antagonism and heterotrimeric G proteins, inhibition of adenylate cyclase, increased Ca^2+^ release and actin polymerization	Inhibition of colonic carcinoma and murine melanoma metastasis	[38,49]
**Neutral effects**			
CXCL1●CXCL7	Activation of CXCR2, Ca^2+^ release, complex interaction with GAGs	CXCR2-transfected HL60 cells response comparable to that with the CXCL7/CXCL1 mixture	[40]
CXCL8●CXCL8	Ca^2+^ mobilization, CXCR1/2 activation and phosphorylation, ERK phosphorylation, desensitization, β-arrestin 1-dependant CXCR2 internalization	CXCR2-induced migration of HMEC and CXCR2-transfected RBL-2H3 cells	[50]
CXCL7●CXCL7	Activation of CXCR2, Ca^2+^ release	CXCR2-transfected HL60 cells response comparable to that with the monomer	[39]
CXCL1●CXCL1	Binding and activation of CXCR2, Ca^2+^ release, ERK phosphorylation	CXCR2-HL60 cells migration comparable to that with the monomer	[51]

## 5. Heterodimerization with Other Soluble Effector Proteins

In addition to chemokine–chemokine interactions, chemokines can bind soluble effectors from other protein families. Here, we describe some examples of those interactions and their functional consequences (Table 1).

### 5.1. Damage-Associated Molecular Pattern (DAMP)

High Mobility Group Box 1 (HMGB1) acts as a DAMP after its release, which can occur passively from dead cells or actively by secretion from different cell types including activated immune cells [93]. In vivo, HMGB1 forms heterocomplexes with CXCL12 [88]. In an air pouch model, the HMGB1•CXCL12 heterodimer promoted leukocyte recruitment to damaged tissue in a CXCR4-dependant but RAGE-independent manner. The synergistic mechanism engaged the β-arrestin proteins into actin polymerization, cell migration, receptor internalization and sorting for degradation [78,89,94]. In addition, computational models revealed that the oxidative state of HMGB1 would regulate the heterodimer complex formation, stressing the fact that the microenvironment greatly influenced the generation of the dimer [95]. HMGB1, which can be massively released within the tumor microenvironment, plays paradoxical roles in cancer depending on its localization [96]. It seems reasonable, but not yet proven, that CXCL12•HMGB1 complexes influence tumor cell migration and metastasis [97].

### 5.2. Antimicrobial Peptides (AMPs)

While HMGB1 has a four-time higher molecular weight than CXCL12, chemokines can also bind smaller proteins. AMPs are naturally occurring small cationic peptides that exhibit potential anti-infective effects and have antimicrobial or immunomodulatory properties. Defensins are AMPs, and there are six known α-defensins in human beings, including human neutrophil peptides (HNPs) 1–4 and human defensins 5–6 [98]. Platelet-derived CCL5 has been shown to form a heterodimer with neutrophil HNP1. In flow chamber assays, pre-incubation of confluent HUVECs with CCL5 and HNP1 resulted in an increase of classical monocyte adhesion [87]. Moreover, in a myocardial ischemia-reperfusion injury model, overexpression of hCCL5 and HNP1 in the myocardium by adeno-associated virus-assisted transduction resulted in enhanced endothelial adhesion of classical monocytes and accumulation of macrophages in the heart.

### 5.3. Evasins

Ticks possess hundreds of bioactive compounds that modulate host defense reactions in their saliva. Among them are Evasins, a class of chemokine-binding proteins comprising three family members: Evasin-1 and Evasin-4 contain six (6-Cys), whereas Evasin-3 has eight (8-Cys) cysteine [99]. Evasin-1 targets CCL3, CCL4 and CCL18; Evasin-3 targets CXCL1 and CXCL8; and Eva-sin-4 targets CCL5 and CCL11. It appears that Evasins could form complexes with chemokines in a mutually exclusive manner, with 8-Cys Evasins binding exclusively to CC chemokines, and 6-Cys Evasins exclusively to CXC chemokines. In a 1:1 stoichiometry, the N-terminal region of CCL3 interacts with both the N-terminal and C-terminal regions of Evasin-1, leading to the formation of three additional secondary structure elements, namely α2 3_10_ helix at the C-terminal of Evasin-1; and the α0 3_10_ helix and β0 strand at the N-terminal region of CCL3. The newly formed structures are aimed to stabilize the complex [100]. Three-dimensional structures of Evasin-3 with CXCL8 have been resolved by solution NMR spectroscopy [101]. Here, the α-helix of CXCL8 is now perpendicular to its β-sheet and all three β-strands were extended upon binding of CXCL8 and two type I turns are created; all of this allows for a higher stabilization of the structure. Tight binding of Evasin-3 to the CXCL8 monomer allowed blocking interactions with CXCR2 and disrupting the GAG-binding interface, leading to the abolition of both chemotactic and haptotactic gradients [101].

### 5.4. Galectins

Cytokines were shown to directly bind β-Galactoside-binding lectins (Galectins, Gal), as exemplified by the formation of a heterodimer between tumor necrosis factor beta and Galectin-2 [102]. Accordingly, dimerization of cytokines with galectins has been referred to as “galectokine” [103]. Galectins are a class of potent cis/trans-acting modulators that function as bridging factors between their carbohydrate recognition domain (CRD) and cell surface glycoconjugates. Galectins have a high affinity for beta-galactose-containing glycoconjugates. Gal-3, a diagnostic marker for risk stratification and prognosis evaluation of heart failure patients, is unique because, in addition to its CRD, it presents a long N-terminal tail (NT), is present in the nucleus and the cytoplasm of cells that may also secrete the molecule, and thereby performs both intracellular and extracellular functions [104,105]. As a proof-of-principle, we showed that Gal-1 and Gal-3 were able to form a heterodimer with numerous chemokines [92]. CXCL12 was able to form a heterodimer with Gal-3 in which Gal-3 binds to CXCL12 via the F-face of its CRD. The opposing glycan-binding S-face and the NT of Gal-3 are not part of the primary interaction domain. CXCL12•Gal-3 interface contains the GAG-binding motif of CXCL12, which primarily involves the first β-strand and 40s residues. This implies that CXCL12 interaction with GAGs is impaired while in silico modelling showed that this heterodimer could bind CXCR4 rather than preventing the chemokine from interacting with its receptor. Thus, the peritoneal recruitment of neutrophils and monocytes in C57BL/6J mice after intraperitoneal injection of CXCL12 was abolished in the presence of Gal-3. Similarly, Sanjurjo et al. showed that CXCL4, CCL5 or CXCL8 could bind Gal-1 and CCL5 with Gal-9. Interaction of CXCL4, but not CCL5 or CXCL8, with the F-face of Gal-1 resulted in CRD structural changes, leading to a higher affinity for specific glycan ligands and glycoproteins [90]. In vitro, the heterocomplex increased the apoptotic rate of Jurkat and phytohemagglutinin-L pre-activated CD3+ and CD3+/CD8+ but not CD3+/CD4+ cells. In contrast, CCL5/Gal-9 heterocomplex inhibited apoptosis in all cell types tested. Therefore, the dimerization of chemokines with galectin might play a role in (chronic) inflammation by regulating galectin activities via modulation of galectin CRD.

## 6. Targeting the Chemokine Interactions

Over the years, protein–protein interactions have gained attention and substantial efforts have been undertaken to identify effective inhibitors. Several approaches could be considered to target chemokines as therapeutic tools for redirecting leukocyte trafficking in inflammatory processes and in chronic diseases.

### 6.1. Small Molecules

Numerous small molecules have been developed to target chemokine receptors. Some of these agents are clinically approved and either block the chemokine-binding pocket of the receptor (e.g., the CXCR4 antagonist plerixafor) or act as non-competitive inhibitors by allosteric binding (e.g., the CCR5 antagonist maraviroc) [106,107,108]. Designing small molecules to block interactions between soluble effector molecules that lack binding pockets and active sites in the same way that receptors and enzymes do is more difficult because the contact area is larger. Nevertheless, the small molecule diflunisal, an NSAID, has been shown to be able to disrupt already formed CXCL12•HMGB1 [109].

### 6.2. Modification of the Primary Structure and Obligate Chemokine Heterodimer

Modifying the target chemokine, in particular at the N-terminus of the protein, which is pivotal for receptor activation, makes for a good antagonist because it would alter the receptor binding of the complex leading to a different cellular response. For example, CCL5 with a remaining N-terminal methionine residue (Met-RANTES) has been shown to bind to CCL5 receptors (CCR1, CCR5) and prevent their activation through the ligands CCL3, CCL4, CCL5. Met-RANTES has been shown to reduce atherosclerosis in Ldlr-deficient mice or to reverse electrocardiac dysfunction (i.e., bradycardia, prolonged PR, and QTc interval) in chronic *Trypanosoma cruzi* infection [110,111]. However, this therapeutic principle does not only prevent the activity of the CXCL4•CCL5 heterodimer, but does block all CCL5-receptor mediated activities resulting in adverse effects such as delayed macrophage-mediated viral clearance and impaired normal T cell functions [72].

In contrast to blocking heterodimer formation, chemokine heterodimers have been tested as therapeutics to treat cancer. The mixture of CXCL4 and CXCL12 or the obligate CXCL4•CXCL12 heterodimer inhibited CXCL12-induced migration of MDA-MB-231 breast cancer cells, increased cytoplasmic Ca^2+^ release and activated downstream signaling of CXCR4 but not CXCR3, highlighting its role in limiting cancer progression. [76,77]. Furthermore, incubation of the chemokine mixture restored the expression of genes that have been associated with a decrease in overall survival in breast cancer patients. In this context, formation of an obligate heterodimer would couple differently to downstream signaling pathways.

### 6.3. Immunomodulation Using Viral Chemokine Binding Proteins (vCKBP)

As mentioned above, pathogens such as viruses or ticks can target different parts of the chemokine network to avoid the host immune response. VCKBP have been shown to inhibit chemokine activity in vitro and in vivo [112,113]. Mechanistically, vCKBP bind to the chemokine through its GAG- and/or chemokine receptor-binding sites, thereby impairing the interaction between the chemokine and its receptor and/or GAGs. The properties of each chemokine binding niche are similar, with mainly hydrophobic residues binding to the N-loop of the chemokine and acidic residues binding to basic chemokine regions, and resemble the interaction of the N-loop with the receptor. Preceding the N-loop is an invariant disulphide bridge, which is common to all four chemokine classes. This structure has been identified as the pivotal and universal recognition site targeted by these soluble viral decoy receptors [114]. Therefore, using such proteins could be of great interest when broad chemokine neutralization is desired. Whether vCKBP affect oligomeric or heteromeric chemokine complexes is unknown.

### 6.4. Synthetic Peptides

Another strategy is to develop synthetic peptides that interfere or mimic the formation of heterodimers [115]. For example, CXCL4 when combined with CXCL12 forms a CXC-type heterodimer that is inhibitory for migration of breast cancer cells. This can be mimicked by AHITSLEVIKAG, a CXCL4-derived peptide that binds CXCL12 [76]. In the case of CXCL4•CCL5, a mouse CCL5-based synthetic cyclic peptide inhibitor called MKEY, which disrupts CCL5•CXCL4, has shown its efficacy by inhibiting atherosclerosis in *Apoe*-deficient mice fed a western-type diet [72]. In a myocardial ischemia and reperfusion model, the disrupting peptide decreased the reduced infarction size by limiting the recruitment of neutrophils and monocytes into the affected myocardial areas [116]. MKEY treatment suppressed the development and progression of abdominal aortic aneurysm in male C57BL/6 mice by attenuating both elastin fragmentation and smooth muscle cell depletion by decreasing the accumulation of macrophages and CD31+ cells in the media and adventitia of the aorta [117]. Finally, injections of MKEY protected against brain injury induced by stroke, mainly by preventing infiltration of circulating monocyte-derived macrophages [118].

CCL5•CCL17 form concentration-dependent, predominantly CC-type heterodimers. The CCL5 derived peptide CAN was designed to prevent the CC-type interaction, which results in functional synergy by promoting heterodimerization of the CCL5 receptor CCR5 and CCR4 [61].

An alternative approach is to mimic the heterodimer by adding either one part or the entire partner. For example, [VREY]4 is a peptide composed of four C-terminal helices of CCL5 (VREY) chemically assembled on a TASP scaffold [61]. [VREY]4 was able to phenocopy the inhibitory effects of CXC-type interactions between CCL5 and CXCL12 and attenuated CXCL12-mediated platelet aggregation. Moreover, a new scaffold version termed i[VREY]4 has been recently developed that has been shown in preclinical models to prevent arterial thrombosis and to improve effects of standard antiplatelet therapy without affecting bleeding [67]. Likewise, evasin-based promiscuous CC-chemokine-binding peptides have also been designed [119]. Those peptides showed high binding affinity, neutralization of chemokine action by preventing receptor binding, and anti-inflammatory activity in vivo.

## 7. Conclusions

Over the last decade, an important accumulation of evidence has shown that the role of (multimeric) chemokines cannot be narrowed to immune cell recruitment and gradient formation only. Chemokines can form interactions with a large number of proteins, even non-protein molecules such as nucleic acids. Those heterocomplexes can be recognized as crosstalk mechanisms with fine-tuning activities leading to new biological functions. Further investigations are needed to elucidate the settings for chemokine dimerization and how chemokine networks regulate and are regulated by the microenvironment under physiological and pathological conditions. Predominantly, the chemokine homodimerization being able to bind to others than their cognate receptor would open new paths for further studies and should be expanded to chemokine heterodimers. In this manner, new insights will help to develop new therapeutic tools.

## Figures and Tables

**Figure 1 ijms-24-10925-f001:**
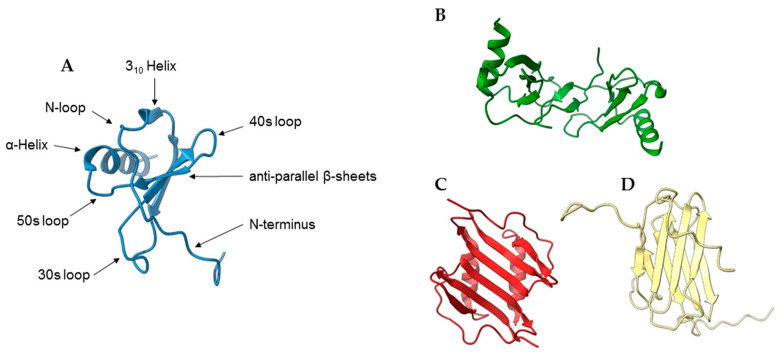
Monomeric and dimeric structures of chemokines. (**A**) monomeric chemokines share a canonical conformation comprising an N-terminal 310-helix, three β-strands and a C-terminal α-helix. (**B**–**D**) CC-type dimers share an elongated feature ((**B**), PDB 2L9H), whereas CXC-type dimers shape into a globular structure with a six-strand-based β-sheet ((**C**), PDB 1RHP). The metamorphic protein XCL1 presents a unique dimeric folding with a four-strand β-sheet ((**D**), PDB 2N54).

**Figure 2 ijms-24-10925-f002:**
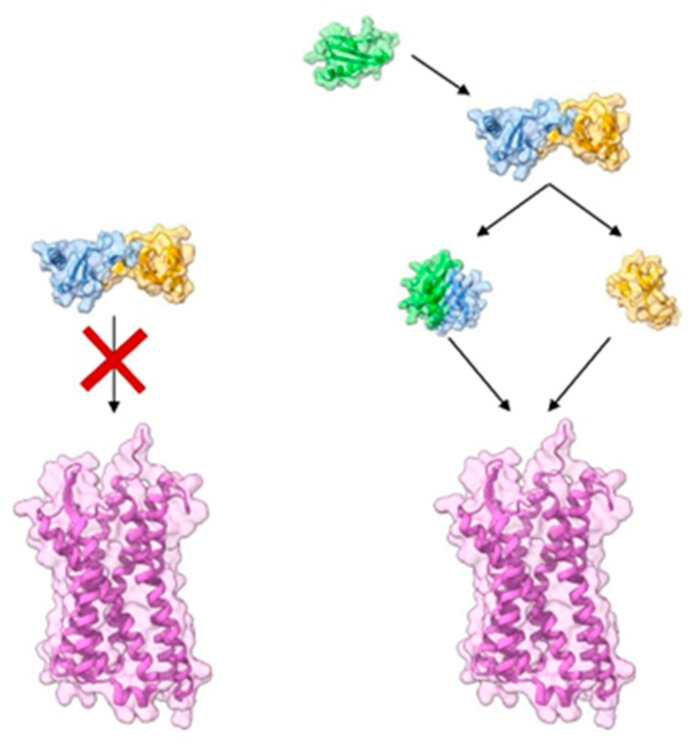
Model of a synergistic switch from a CC-type to a CXC-type dimer. (**Left**): CC-type CCL2 homodimer (blue and yellow, PDB 1DOL) is unable to bind CCR2 (purple, PDB 7XA3) due to the overlapping chemokine regions involved in both homodimeric formation and receptor binding. (**Right**): a chemokine (e.g., CXCL4 in green, PDB 1RHP) competes and displaces the dimer leading to the generation of a new homodimer (CXC-type in blue and green) with the release of free monomeric CCL2 (yellow). In this case, both monomeric CCL2 and the CXC-type homodimer are potentially able to bind to CCR2.

## Data Availability

Not applicable.

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
