# Peer review of "Chemokine Heteromers and Their Impact on Cellular Function—A Conceptual Framework"

_ijms, 2023, doi:10.3390/ijms241310925_

Round 1
Reviewer 1 Report
In the current review, Blanchet and colleagues gave a good perspective on the current knowledge about chemokines dimerization and their impact on cellular function. I suggest to the authors expand their research interest in the molecular dynamics simulations approaches for unveiling how environmental conditions affect chemokines multimerization. Ultimately, this will lead also pave the way for the development of new medical treatments.
Author Response
We thank the reviewer for the positive comment and directing us to improve the manuscript. We included Sepuru et al., Biochem J, 2017, now reference 65 one important paper that used MD simulations to investigate GAG chemokine interactions using non-dissociating CXCL1•CXCL2. Also the finding from Liang et al. that GAGs bind different to monomers and higher order oligomers fits to this paragraph.
Line 265-267: Molecular docking based on NMR experiments combining the obligate heterodimer CXCL1•CXCL2 shows that asymmetry caused by chemokine heterodimerization affects binding to GAGs and that it matters whether GAGs are soluble or immobilized (65).
Page 4, lines 175-181: Furthermore, structure experiments reveal that oligomerization of chemokines affects how GAGs bind chemokines. CCL3, CCL4 and CCL5 have a common and pivotal GAG binding motif in the 40s loop that becomes partially buried at the moment these CC chemokines associate to higher order oligomers. While the CCL5 polymer binds GAGs through an additional motif in the 50s loops [35], CCL3 polymers allow GAG binding through novel binding sites that develop by proximity of two partially buried BBXB motifs [36].
Reviewer 2 Report
This manuscript is a well written important review about the role and impact of chemokine heteromers including various types of diseases.
The manuscript entitled "Chemokine Heteromers and Their Impact on Cellular Function" is a review that focuses on the function of chemokines and the consequences of their interactions with other proteins. The authors also explore recent potential therapeutic tools targeting the multimeric structures of chemokines, which can modulate their biological functions. The topic of this review is original, as it covers the involvement of chemokines in various biological processes under physiological conditions, including homeostasis, development, tissue repair, and angiogenesis. Furthermore, it addresses their roles in pathological disorders such as tissue damage, bacterial infection, tumorigenesis, cancer metastasis, inflammatory- and autoimmune diseases.
The Figures are well designed.
However, the manuscript could benefit from an update in its references, particularly regarding different malignancies."
Author Response
We thank the reviewer for the constructive suggestion to add more information about chemokine heteromers and oligomers in malignancy. This is an important topic. Not many papers exist about this subject. We added now a review by D’Agostino et al. (ref.80), and added a paragraph of heteromers in malignancy to highlight this topic:
Page 7, lines 322-329: Because the tumor microenvironment can express multiple soluble factors including chemokines, it is highly likely that heterocomplexes can be formed as well [80]. For instance, CXCL9 and CXCL12 are expressed in the perivascular space of primary central system lymphoma that consists mainly of CD8 T cells. Co-existence of both chemokines resulted in CXCL9•CXCL12 complexes and enhanced recruitment of B cells that lack the CXCL9/10/11 receptor CXCR3. To exploit this mechanism therapeutically and provide proof of causality specific inhibitors of the CXCL12•CXCL9 interaction would be beneficial [81].
Page 12 line 413-416: HMGB1, which can be massively released within the tumor microenvironment plays paradoxical roles in cancer, depending on its localization [98]. It seems reasonable, but not yet proven, that CXCL12•HMGB1 complexes influence tumour cell migration and metastasis [99].
Reviewer 3 Report
In this review, authors make a strong case for the importance of chemokine heteromers and how interactome studies can completely miss heteromer formation for soluble proteins unlike for membrane and intracellular proteins. They make a case of how heteromer formation for other important mediators such as cytokines and G protein ligands and by extension that chemokine heteromers play an important role in human pathophysiology and could be novel drug targets.
1) However, the review could be better organized by paying more importance on how heterodimer formation and receptor and GAG interactions promote in vivo function. For instance, the authors rightly mention upfront in the abstract about the heterodimer-GAG interactions, but the main text has no description of GAG interactions of heteromers. This reviewer is aware of at least two studies that have directly addressed GAG-heteromer interactions. Ref. 39 discusses GAG binding of the CXCL7-CXCL1 heterodimer, and a recent study describes GAG interactions in great detail of the mouse CXCL1-CXCL2 heterodimer (PMID: 33463672).
2) Authors cite a large number of studies and it will be helpful if the authors critically assess these data by explaining whether the phenotype observed can be attributed to heterodimers and not due to synergy and non-additive interactions.
3) Introductory sections are descriptive and could be trimmed to make it more cogent and germane to the theme of the review – namely heterodimers and function.
For example, the following paragraphs and sentences could be removed or modified to make it relevant to the review’s theme.
Mantovani et al. proposed another classification of chemokines based on their expression and functional activity [11].----
The human genome encompasses more than 50 different chemokine genes and pseudogenes. These genes have undergone a rapid evolution in both their sequences and their family gene size [10].
Although chemokines folds into a canonical tertiary structure, some can display specificities. For instance, CCL28, also known as mucosae-associated epithelial chemokine or MEC, is a chemokine -----
4) Besides CXCL4, CXCL7 also exists as monomers, dimers, and tetramers, and its oligomerization is sensitive to pH and buffer conditions (PMID: 28245630; PMID: 8051099). It is interesting that both are platelet-derived chemokines, present in large amounts, and form heterodimers. Authors should comment on the relevance of these observations for function.
5) Authors mention that CXCL4 bound to CXCL8 with high affinity and cite refs. 74, 75. Ref. 74 is a functional study and ref. 75 is a computational docking study. However, in ref.38, Brown et al show that CXCL4 does not interact with CXCL8 and propose electrostatic repulsion disfavors heterodimer formation. The authors need to reconcile these differences and discuss whether the observations in ref. 74 is not due to heterodimer formation but to synergistic interactions.
6) Authors describe one of the models for heterodimer signaling as a heterodimer binding to a heterodimer chemokine receptor. I presume that the authors imply that this signaling event is novel and different from those of the monomers and homodimers. This is interesting, and the authors should expand on this section as it will help the readers better appreciate this novel mechanism. All of the chemokine-receptor structures at this time show either a chemokine monomer or dimer bound to a single receptor. Further, for several CXC chemokines, only one monomer is bound to the receptor with the second monomer pointed away and making no interactions with the receptor.
7) under chemokine heterodimerization activity, Ref. 38 shows that the activity of the heterodimer is no different from CXCL1 and CXCL7 must be cited and described. It will add value to this review if the authors provide a mechanism(s) for how different heterodimers regulate in vivo function. At this time, authors have described different studies but have not provided a comprehensive mechanism.
Author Response
1) However, the review could be better organized by paying more importance on how heterodimer formation and receptor and GAG interactions promote in vivo function. For instance, the authors rightly mention upfront in the abstract about the heterodimer-GAG interactions, but the main text has no description of GAG interactions of heteromers. This reviewer is aware of at least two studies that have directly addressed GAG-heteromer interactions. Ref. 39 discusses GAG binding of the CXCL7-CXCL1 heterodimer, and a recent study describes GAG interactions in great detail of the mouse CXCL1-CXCL2 heterodimer (PMID: 33463672_no access).
We agree with the third reviewer on the importance of heterodimer formation and receptor and GAG interactions. We updated our manuscript with the following section:
Page 7 line 330-338: Using a trapped/obligate CXCL1•CXCL7 heterodimer, Brown et al. showed that the heterocomplex residues involved in GAG binding were different from their respective monomers. As for monomeric CXCL1 and CXCL7, heterodimer-GAG interactions blocked the residues engaged in receptor binding, indicating that GAG-bound CXCL1•CXCL7 is unlikely to activate the receptor. Furthermore, CXCR2 receptor binding activities showed no differences between CXCL1, CXCL7, a mixture of both chemokines (i.e. native heterodimer) and the trapped heterodimer. Therefore, receptor activity and functional activities could be spatially and temporally controlled by GAGs through modulation of the amount of free chemokines.
2) Authors cite a large number of studies and it will be helpful if the authors critically assess these data by explaining whether the phenotype observed can be attributed to heterodimers and not due to synergy and non-additive interactions.
We added a pargraph about this:
Page 9, lines 375-390: Although numerous studies have highlighted the role of chemokine heterodimers in cellular function, some might present limitations. The major critical aspect is the use of individual chemokines. Indeed, the combination of different concentrations of chemokines will result in a solution containing a mixture of different oligomeric states, with monomeric, homo-/heterodimeric and multimeric structures. Whether the observed amplified or synergistic effects could also explain intracellular effects at the signaling level remains speculative. Trapped chemokine heterocomplexes appear to be the most prominent solution to overcome the issues as only one entity is present in solution and its concentration can be fine-tuned to accurately control the experimental conditions. However, the construction of such proteins requires structural data that are often not available, rendering its design challenging. So far, two methods have been used to generate obligate/trapped heterodimers according to structural models derived from HSQC spectra and MD simulations. CXC-type heterodimers have been generated by mutation of opposite relevant residues at the interface to cyteines and subsequent disulfide bridge formation [39, 65]. An alternative approach to create obligate CC-type heterodimers has been chemical synthesis with oxime-ligation of opposite residues in the N-terminal region [86]
3) Introductory sections are descriptive and could be trimmed to make it more cogent and germane to the theme of the review – namely heterodimers and function.
For example, the following paragraphs and sentences could be removed or modified to make it relevant to the review’s theme.
Mantovani et al. proposed another classification of chemokines based on their expression and functional activity [11].----
The human genome encompasses more than 50 different chemokine genes and pseudogenes. These genes have undergone a rapid evolution in both their sequences and their family gene size [10].
Although chemokines folds into a canonical tertiary structure, some can display specificities. For instance, CCL28, also known as mucosae-associated epithelial chemokine or MEC, is a chemokine -----
We agree that the removal of the mentioned sentences would improve our manuscript.
4) Besides CXCL4, CXCL7 also exists as monomers, dimers, and tetramers, and its oligomerization is sensitive to pH and buffer conditions (PMID: 28245630; PMID: 8051099). It is interesting that both are platelet-derived chemokines, present in large amounts, and form heterodimers. Authors should comment on the relevance of these observations for function.
We thank the reviewer for pointing out that these two important papers, that are now included as references 23 and 24 deserve to be discussed as they provide guidance on the influence of concentration and buffer conditions on CXCL7 oligomerization (Yang et al., JBC 1994) and the importance of the CXCL7/NAP-2 monomer, respectively (Brown et al., IJMS 2017).
The following section has been added to the manuscript
page4 line130-143: Oligomerization of CXCL7, like CXCL4 most abundantly expressed by platelets and released during platelet activation, depends on chemical environment (Brown et al., IJMS 2017 a, Yang et al., JBC 1994). Both chemokines have been shown to exist as monomers, dimers and tetramers. Increasing the pH from 3.5 to 6 increases the association equilibrium for dimer formation of all CXCL7 species depending on the length of the N-terminus (PPBP> CTAPIII> β-TG >NAP-2) while that of the association to tetramers is decreased (Brown et al., IJMS 2017, Yang et al., JBC 1994).). Ionic strength plays also a role but effects are smaller and are not linear. Effects are due to modification of the global charge of some residues leading to the disruption of intramolecular electrostatic interactions. On the other hand, these modifications allow the generation of new electrostatic interactions with other proteins or the formation of salt bridges with ions from the milieu (Yang et al?). Then, these changes would foster the formation of new oligomeric states, which would promote or alter the chemokine gradient formation, the generation of new heterodimer and influence the receptor binding activity (e.g. CXCR2) which would modulate the cellular/immune response eventually (e.g. neutrophil recruitment to the site of injury.
Page 5 line 188-193: Investigating heparin binding of NAP-2/CXCL7 Brown et al. showed that heparin exerts effects on multiple NAP-2/CXCL7 residues so that not one single binding model can harmonize all chemical shift perturbations (Brown structural basis of native CXCL7 monomer) indicating that the GAG-interface is plastic. In addition, GAG-binding sites appear to overlap with effects of the CXCR2 N-terminus so that Nap-2/CXCL7 monomers bound to GAGs would not be able to fully bind CXCR2.
5) Authors mention that CXCL4 bound to CXCL8 with high affinity and cite refs. 74, 75. Ref. 74 is a functional study and ref. 75 is a computational docking study. However, in ref.38, Brown et al show that CXCL4 does not interact with CXCL8 and propose electrostatic repulsion disfavors heterodimer formation. The authors need to reconcile these differences and discuss whether the observations in ref. 74 is not due to heterodimer formation but to synergistic interactions.
In reference 74, Dudek et al. show binding of CXCL8 to immobilized CXCL4 on a Biacore system, CSP of N15-CXCL4 and Nesmelova et al. show by mass spectrometry an interaction. The evidence is compelling. In the mentioned reference from Brown et al. in IJMS 2017 no experiments are described that investigate CXCL8 binding to CXCL4. But, in general depending on the method it is possible that existing interactions can be not detectable.
6) Authors describe one of the models for heterodimer signaling as a heterodimer binding to a heterodimer chemokine receptor. I presume that the authors imply that this signaling event is novel and different from those of the monomers and homodimers. This is interesting, and the authors should expand on this section as it will help the readers better appreciate this novel mechanism. All of the chemokine-receptor structures at this time show either a chemokine monomer or dimer bound to a single receptor. Further, for several CXC chemokines, only one monomer is bound to the receptor with the second monomer pointed away and making no interactions with the receptor.
We agree with the reviewer. It was our purpose to suggest a novel mechanism how chemokine activity might be regulated if two distinct chemokine are combined. Association of two CC-chemokines to CC-type homodimers can lead to inhibitory effects if both N-termini that are important for receptor activation are prevented from entering the receptor pocket. The addition of another chemokine that results in competitive interaction with the homodimer would a) result in increasing concentrations of the CC chemokine monomers because the homodimer disassembles and b) result in a switch to new heterodimers of the CXC-type species which could bind and activate the receptor. The contact sites of the chemokine heterodimer to the receptor do not need to be specified in this concept.
We changed the caption of Figure 2 into: Model of a synergistic switch from a CC-type to a CXC-type dimer
7) under chemokine heterodimerization activity, Ref. 38 shows that the activity of the heterodimer is no different from CXCL1 and CXCL7 must be cited and described. It will add value to this review if the authors provide a mechanism(s) for how different heterodimers regulate in vivo function. At this time, authors have described different studies but have not provided a comprehensive mechanism.
We agree with the reviewer that “Brown, A. et al. Platelet-derived chemokine CXCL7 dimer preferentially exists in the glycosaminoglycan-bound form: Implications for neutrophil–platelet crosstalk. Frontiers in Immunology 2017”, now reference 35 is an important paper (and we cite it at several occasions page 5 line 1, page 8 line 357). It is one of the few examples using a non-dissociating "trapped" homodimer of Nap-2/CXCL7 to study its function. Notably, the CXCL7 homodimer is functionally active and induces calcium release in CXCR2 transfectants. However, because the functional activities of chemokines are diverse and multiple signaling pathways are activated by CXCL7-CXCR2, it is very difficult to conclude that the activity of the different oligomeric states of CXCL7, especially between monomers and dimers, is not different.
There appears to be not one single comprehensive mechanism to explain the effects that can be synergistic or inhibitory. It all depends on the respective chemokine pairs that form complexes which is explained at the respective positions in the manuscript. To take this into account and that the models we propose are concepts we modified the title: Chemokine heteromers and their impact on cellular function – a conceptual framework.
In addition, we provide a new table summarizing the mechanisms how chemokine complexes affect cellular function
Round 2
Reviewer 3 Report
the revised manuscript can now be accepted